# Study of Gold Nanoparticles Conjugated with SARS-CoV-2 S1 Spike Protein Fragments

**DOI:** 10.3390/nano13152160

**Published:** 2023-07-25

**Authors:** Žiga Jelen, Janez Kovač, Rebeka Rudolf

**Affiliations:** 1Faculty of Mechanical Engineering, University of Maribor, Smetanova ulica 17, SI-2000 Maribor, Slovenia; z.jelen@um.si; 2Department of Surface Engineering, Institut Jožef Stefan, Jamova 39, SI-1000 Ljubljana, Slovenia; janez.kovac@ijs.si

**Keywords:** gold nanoparticles, ultrasonic spray pyrolysis, conjugation, SARS-CoV-2 S1, characterisation

## Abstract

This study reports on the successful conjugation of SARS-CoV-2 S1 spike protein fragments with gold nanoparticles (AuNPs) that were synthesised with Ultrasonic Spray Pyrolysis (USP). This method enables the continuous synthesis of AuNPs with a high degree of purity, round shapes, and the formation of a surface that allows various modifications. The conjugation mechanism of USP synthesized AuNPs with SARS-CoV-2 S1 spike protein fragments was investigated. A gel electrophoresis experiment confirmed the successful conjugation of AuNPs with SARS-CoV-2 S1 fragments indirectly. X-ray Photoelectron Spectroscopy (XPS) analysis confirmed the presence of characteristic O1s and N1s peaks, which indicated that specific binding between AuNPs and SARS-CoV-2 S1 spike protein fragments takes place via a peptide bond formed with the citrate stabiliser. This bond is coordinated to the AuNP’s surface and the N-terminals of the protein, with the conjugate displaying the expected response within a prototype LFIA test. This study will help in better understanding the behaviour of AuNPs synthesised with USP and their potential use as sensors in colorimetric or electrochemical sensors and LFIA tests.

## 1. Introduction

Conjugation, or the binding of specific molecules or functional groups to the surface of nanoparticles [1,2,3], is one of the most useful chemical methods of modifying the surface properties of nanoparticles. Gold nanoparticles (AuNPs) represent one of the most common conjugation targets. This is mainly due to the surface state of AuNPs, which enables specific and non-specific binding. At the same time, AuNPs have a high degree of stability, which results in inertness and high biocompatibility. Various synthesis techniques allow us to achieve the required sizes and shapes of AuNPs [4,5], with which we can affect their most important property directly, i.e., surface plasmon resonance in the visible spectrum of light [6,7]. Due to the presented special properties, AuNPs have found uses in drug delivery for cancer treatment [8,9], in imaging as contrast agents [10,11], as markers in diagnostic tests [12,13], and as colorimetric [14,15] and electrochemical sensors [16,17,18]. The listed sensors, based on AuNPs, are most widespread in the form of (i) target-mediated aggregation, where stable dispersions of functionalised AuNPs aggregate in the presence of target analytes, (ii) substrate-mediated catalytic activity, where the enzyme-like activity of AuNPs is utilised to enhance the sensitivity and selectivity of sensing systems, and (iii) point-of-care testing sensors in the form of microfluidic chips and lateral flow assays (LFIA) [13,14]. LFIA tests are based on the use of the surface plasmon resonance properties of conjugated AuNPs, where they act as coloured labels that appear as test lines in the visible spectrum. AuNPs, which are used as starting raw materials for the preparation of labels, must have high stability in dry conditions, good dispersibility, and high colloidal stability in suspension under different conditions, as well as good redispersion. Conjugation with AuNPs should be reproducible and stable.

The outbreak of the COVID-19 pandemic has moved LFIA diagnostic devices based on the use of AuNPs to the forefront as the first diagnostic method in diagnosing SARS-CoV-2 virus infection [19,20,21]. Their use has increased tremendously due to relatively simple production and good commercial availability [22,23]. For the purposes of LFIA tests, AuNPs are synthesised using various chemical and physical methods, each of which has its own advantages and disadvantages [24,25]. Most AuNP synthesis methods have low productivity (few mg/h). Until now, two key methods have been established at the industrial level: the reduction of Au^3+^ and Au^1+^ solutions [25,26] and laser ablation [27,28,29], which enable synthesis only in limited quantities [30]. Previous studies of the physical properties of AuNPs have shown that a size between 40 nm and 80 nm and a high degree of sphericity (>0.97) enable the achievement of higher accuracy and sensitivity [31] of the LFIA tests. Studies of the surface chemical properties of AuNPs showed that these are mainly dependant on the method of synthesis [32,33], and the conjugation of molecules to nanoparticles involves chemically linking or adsorbing the molecules onto the nanoparticle’s surface. The choice of conjugation strategy depends on the nature of the nanoparticles and the desired interaction with the target system [1,30,32].

In this study, we investigated the conjugation mechanism of AuNPs synthesised using Ultrasonic Spray Pyrolysis (USP), which, compared to other established methods, enables the continuous and easily scalable synthesis of large quantities of AuNPs with precise control over particle size and morphology, resulting in a uniform distribution of nanoparticles with a high degree of purity, round shapes, and with a Au surface [34], which gives the possibility of various AuNP surface modifications. The technique can be used to produce a wide range of materials, including metal oxides, semiconductors, and ceramics, while also allowing for controlled doping with various elements, enabling the preparation of tailored nanoparticles with functional properties.

In the USP synthesis of AuNPs, the nanoparticles are collected in deionised water; consequently, it is necessary to dry the resulting suspension, for certain use cases, which also simplifies the subsequent manipulation of dried AuNPs during the conjugation process.

Based on the above, it was hypothesised that USP could have the potential to become one of the leading methods of AuNP synthesis for AuNP-based sensors, such as the labels used in LFIA tests. A study of the conjugation of AuNPs with SARS-CoV-2 S1 spike protein fragments was performed with the aim of confirming the presented hypothesis. Comparative analysis methods were used with the aim of confirming that AuNPs synthesised using USP have an adequate surface for conjugation with specific molecules.

## 2. Materials and Methods

### 2.1. Synthesis of AuNPs

AuNPs were synthesised from an aqueous solution of gold chloride (hydrogen tetrachloroaurate (III) trihydrate, HAuCl_4_, Glentham Life Sciences, Corsham, UK) with a concentration of 1.0 g/L. The custom USP device, located in Zlatarna Celje d.o.o. (Celje, Slovenia), consists of a 1.65 MHz piezoelectric aerosol generator and a furnace with three heating zones (one evaporation and two reaction)—Figure 1. The aerosol was formed in the generator and transported with N_2_ carrier gas through the evaporation zone (T = 200 °C) into the reaction zones (both with T = 400 °C), where H_2_ was introduced as a reducing gas, which enabled the reduction of the gold salt into elemental gold. The synthesised AuNPs were collected in gas wash bottles, containing only deionised water or deionized water with 2.5 g/L polyvinylpyrrolidone (PVP, average M.W. 40,000, Sigma-Aldrich, China), in order to prevent agglomeration. Surface modification of PVP-stabilised AuNPs was conducted immediately after USP synthesis with the addition of 1 g/L of trisodium citrate (Merck KGaA, Darmstadt, Germany), while 1 g/L of sucrose (Merck KGaA, Darmstadt, Germany) was added to improve the stability during freeze drying.

### 2.2. Drying

Freeze-drying was performed on a Labfreez instruments FD-200F SERIES (Labfreez instruments group Co., Ltd., Beijing, China), over the course of 49 h. The freeze-drying parameters were set as follows: (1) 4 h for initial freezing of the AuNP suspension at ambient pressure, with a shelf temperature of −40 °C; (2) 12 h primary drying at a pressure of 1–4 Pa and a shelf temperature of +20 °C, and (3) 33 h of primary and secondary drying until the product was completely dry, at a pressure of 1–4 Pa and a shelf temperature of +30 °C.

### 2.3. Preparation of the AuNP Suspension

Freeze-dried AuNPs were redispersed in a phosphate buffer solution (PBS) and sonicated for 30 s. The dispersion was subsequently centrifuged at 10,000 rpm and 4 °C for 40 min; 90% of the remaining supernatant was removed, discarded, and replaced with 40% of the initial volume of fresh PBS.

### 2.4. Conjugation

The conjugation of USP-synthesised AuNPs was conducted by adding 100 µg of SARS-CoV-2 S1 spike protein fragment aa 1-681 (EMD Millipore, Temecula, CA, USA) to 1 mL of purified USP AuNPs, with an optical density (OD) of approximately 3. The mixture was mixed and incubated at room temperature for 30 min, followed by the addition of 0.1 mL of a 10 mg/mL BSA in PBS solution to block the surface. The resulting conjugate suspension was centrifuged for 20 min, at 10,000 rpm, at 4 °C. Afterwards, the supernatant was removed and replaced with 1 mL of a 1 mg/mL BSA in PBS solution. The new solution was sonicated for 10 s to mix the compacted conjugate. These steps were repeated 3 times. After the fourth and final cleaning, the removed supernatant was replaced with 1 mL of PBS. A schematic representation of the AuNP conjugation process is presented in Figure 2.

### 2.5. AuNP Characterisation

#### 2.5.1. TEM Microscopy

The morphology of the AuNPs was investigated by means of a JEOL 2100 FEG transmission electron microscope (TEM) (Jeol, Japan). The TEM investigations were carried out at 200 kV of accelerating voltage. The AuNPs were suspended in absolute anhydrous ethanol and mixed in an ultrasonic bath for 5 min. Then, a few drops of the suspension were pipetted onto a TEM grid coated with holey carbon films (Ted Pella Inc., Redding, CA, USA) and air-dried.

#### 2.5.2. Zeta Potential

Unmodified AuNPs, with no added stabilisers or surface modifiers, were characterised by zeta potential measurements using a Malvern Zeta sizer (Malvern Panalytical, Worcestershire, UK). For the measurement, HCl was added to the AuNP suspension, the pH was lowered to 1, and the measuring cell of the instrument was then filled with this suspension. The material property parameters used were as follows: absorption 0.010, refractive index 1.59, dispersion properties of water, temperature 25 °C, and measurement angle: 173° back scatter.

### 2.6. AuNP Conjugate Characterisation

#### 2.6.1. UV-Vis

Ultraviolet–visible spectroscopy (UV-Vis) measurements were performed with a Tecan Infinite M200 UV/Vis spectrophotometer (Tecan Group Ltd., Männedorf, Switzerland), using a special microplate, with the following parameters: sample volume: 300 µL, absorbance range: λ = 400 to 700 nm, and no. flashes: 5×.

#### 2.6.2. Gel Electrophoresis

Gel electrophoresis experiments were performed with a PhastSystem (GE Healthcare Bio-Sciences AB, Uppsala, Sweden) electrophoresis device, in a standard 0.3% agarose gel with a 125 × TAE buffer solution at an electrical differential of 180 V. The experiments were run for 15 min. In addition, 1 % Sodium dodecyl sulfate (SDS) was added to selected samples, which enabled the reliable separation of unconjugated and conjugated AuNPs within the method.

#### 2.6.3. XPS Spectroscopy

In the X-ray Photoelectron Spectroscopy (XPS) characterisation, the focus was on three target samples: (1) stabilised AuNPs (Sp1), (2) purified AuNPs prepared for conjugation (Sp2), and (3) conjugated AuNPs (Sp3). Table 1 shows the composition of each sample.

The measurements were carried out with a PHI TFA XPS spectrometer (Physical Electronics, Chanhassen, MN, USA) equipped with an Al anode providing a monochromatic X-ray source with an energy of 1486.6 eV. A drop of the sample suspension was placed on a Si substrate and dried in a desiccator for 24 h. The energy scale of the XPS spectra was calibrated by the carbon C 1s peak at 284.8 eV. The analysis depth of the XPS method is 3–5 nm. The energy resolution of the XPS spectra was about 0.65 eV. The deconvolution of the XPS spectra and calculation of the chemical composition were performed with MultiPak software, ver. 9.9, provided by Physical Electronics. Samples for XPS investigations were prepared by depositing a single drop onto a Si substrate and then drying.

### 2.7. Preparation of LFIA Test Strips

To confirm the functionality of the conjugated AuNPs, experimental LFIA test strips were prepared for the detection of SARS-CoV-2 S1 IgA, IgG, and IgM antibodies. For the test strips, we used a cellulose sample and absorption pad (CFSP173000, EMD Millipore, CA, USA), a pretreated polyester conjugate pad (6614, Ahlstrom-Munksjö, WI; USA), a plasma separation pad (Cytostep 1668 (HV+), Ahlstrom-Munksjö, USA), and a Hi-Flow™ Plus 180 nitrocellulose membrane (EMD Millipore, USA). The test and control lines were applied using a small fine-tipped silicone brush (Koi Water Brush #2, Sakura Color Products Corp., Osaka, Japan) and protein solutions with a concentration of 1 µg/mL for a final test line concentration of 0.1 µL/mm. The test lines were spaced 3 mm apart from each other. Anti-Human IgA (α-chain specific) antibodies (Sigma-Aldrich, Saint-Louis, MO, USA) were used to detect IgA antibodies for SARS-CoV-2 S1. Anti-Human IgG (Fab-specific) antibodies (Sigma-Aldrich, Saint-Louis, MO, USA) were used to detect IgG antibodies for SARS-CoV-2 S1. Anti-Human IgM (μ-chain specific) antibodies (Sigma-Aldrich, Saint-Louis, MO, USA) were used to detect IgM antibodies for SARS-CoV-2 S1, and Anti-Human IgG (Fc specific) antibodies produced in rabbits (Sigma-Aldrich, Saint-Louis, MO, USA) were used for the control line, in combination with AuNPs conjugated with anti-rabbit IgG antibodies, goat monoclonal (SigmaAldrich Chemie GmbH, Darmstadt, Germany). The assembled test sheets (300 mm wide) were cut into 4 mm wide test strips.

## 3. Results and Discussion

### 3.1. USP Synthesis and Drying

The AuNPs synthesised from an aerosolised aqueous solution of tetrachloroaurate(III) in the reaction zone of the USP device were most likely formed by a Droplet-To-Particle (DTP) mechanism, as previously reported by Majerič et al. [35,36]. The DTP mechanism is based on the formation of a single nanoparticle from a single aerosol droplet through the evaporation of the solvent, and later, as a dry particle, it is subjected to chemical reactions of the remaining solute substance. By changing the initial concentration of the solute in the precursor, it is possible to exercise precise control over the final size of the synthesised nanoparticles. As a general rule, lower solute concentration results in smaller and rounder nanoparticles. The synthesised nanoparticles are collected in a different collection system of the USP device [37,38]. In this study, AuNP collection in demineralised water with a selected PVP stabiliser was used, which proved to be very effective. For subsequent manipulation of the resulting AuNPs, it was necessary to remove the excess water, for which freeze-drying was used, where the excess water was removed by sublimation at reduced pressures. It is a relatively mild drying process, which helps prevent the agglomeration of the AuNPs. In order to reduce the additional risk of agglomeration of AuNPs during the freezing stage, cryoprotectants such as trisodium citrate and sucrose [39] were added for the freezing phase. The resulting dried AuNP cakes were used for the conjugation process.

### 3.2. AuNP Characterisation

#### 3.2.1. TEM Results

The TEM investigations revealed that the AuNPs were round and that their sizes were below 80 nm, as can be seen from Figure 3a. This directly confirms the statement that AuNPs were formed by the DTP mechanism. Figure 3b provides a detailed view of the crystal lattice on the surface of the AuNPs. By comparing the measured distance between the crystal planes, which was 0.40786 nm, with the theoretical distance between the crystal planes in cubic face-centred gold, which was 0.40782 nm, the presence of cubic gold can be confirmed as Fm-3m on the AuNP’s surface. This information confirms the formation of elemental gold directly.

The synthesised AuNPs were also in accordance with the general recommendations for nanoparticle sizes that can later be used for labels, in order to achieve the requirements for good sensitivity and effectiveness in LFIA tests [12,31,40]. Considering this, a more precise analysis was conducted of the size distribution of the AuNPs. A size analysis of 10 TEM images with 250 AuNPs showed a distribution where the majority of AuNPs were between 45 and 55 nm (Figure 4).

#### 3.2.2. Zeta Potential

The zeta potential measurements showed that the initial unmodified AuNPs had a very low surface charge before stabilisation or conjugation (Figure 5), indicating their significant tendency to agglomerate. This low charge required the use of stabilisers; therefore, PVP was used in this study, the addition of which increased the stability of the AuNP suspensions significantly. Due to the use of PBS (pH 7.5), the subsequent zeta potential measurements were performed at a pH of 7.5, where the addition of PVP only influenced the potential slightly, with a shift from −2.4 mV to −3.2 mV.

### 3.3. AuNP Conjugate Characterisation

#### 3.3.1. UV-Vis

The AuNP suspensions were evaluated with the UV-Vis method before and after conjugation. We detected a shift in the peak maximum of the absorption spectrum, as can be seen in Figure 6, from 532 nm for the AuNP suspension to 540 nm for the AuNPs conjugated with SARS-CoV-2 spike protein fragments. The characteristic peak shift to higher wavelengths is indicative of the successful conjugation [41]. The lower peak intensity of the unconjugated AuNPs and conjugated AuNPs was due to losses of unconjugated AuNPs during the centrifugation and purification steps.

#### 3.3.2. Gel Electrophoresis

With the aim of confirming the conjugation of AuNPs with SARS-CoV-2 S1 spike protein fragments, target samples from suspensions of unconjugated and conjugated AuNPs were prepared as shown in Table 2.

Since the previous measurements of the zeta potential showed an almost neutral charge of AuNPs, the test samples of suspensions (2, 4, and 6) were charged additionally with the addition of 1% SDS [42], which usually results in an excess of negative charge. In samples 3 and 4, the pH of the suspension was stabilised by the addition of disodium phosphate, which represents an additional buffer. Samples 5 and 6 represented a conjugate of AuNPs and SARS-CoV-2 S1 fragments. As can be seen in Figure 7, after t = 15 min of exposure to an electrical differential of 180 V (arrow 1), in the case of the unmodified AuNPs, they did not travel from their initial pockets. The slight change in colour was due to AuNP diffusion from the sample into the surrounding gel. This result confirms the neutral charge of AuNPs indirectly and additionally. In samples 2 and 4, where SDS was added [42], the AuNPs travelled towards the anode, which can be seen clearly in Figure 7 (areas marked with a and b). In the case of the conjugated AuNPs, the addition of SDS resulted in a significantly longer path, as can be seen in the area marked c (Figure 7). We concluded that this was the result of a higher charge of the conjugated AuNPs, since the proteins bound to their surface enabled a greater number of binding sites for SDS, and consequently, created a higher charge.

#### 3.3.3. XPS Characterisation

In the XPS investigations, the aim was to compare three different types of samples: stabilised AuNPs, purified AuNPs that were ready for conjugation, and conjugated AuNPs. The goal was to detect any change in the chemical composition right on the AuNPs’ surfaces. This could represent an additional confirmation for the successful conjugation.

The measured chemical composition of the surfaces of the selected samples is shown in Table 3. Since PBS was used in all samples, this was reflected in the presence of Na, P, and Cl in the XPS spectra. The presence of Si was due to the signal from the sample holder.

Monitoring the C 1s, N 1s, O 1s, and Au 4f spectra was key to understanding the conjugation mechanism between the AuNPs and SARS-CoV-2 S1 fragments. In the case of the C 1s spectra (Figure 8a) for unpurified AuNPs, they showed a peak at 284.8 eV. This peak was assigned to the C-C/C-H bonds of the carbon atoms (284.8 eV) [43], which are often detected on samples during XPS analysis. The O 1s spectra of unpurified and purified AuNPs showed a prominent peak at 531.4 eV, which may be assigned to oxygen (C=O) bound in the PVP and oxygen (C=O, C-O-H) bound in the citrate [44,45]. In the case of the conjugated AuNPs, a shift of the peak to a lower energy of 530.8 eV was visible, which indicated an increased proportion of C-O-H bonds [45].

Figure 9a shows a stack of XPS spectra for N 1s. In the case of the unpurified sample, a distinct sharp peak at 399.4 eV can be observed, which is consistent with the nitrogen signal in PVP (399.9 eV [44,46]). The intensity and sharpness of the peak indicated that this sample was dominated by the PVP. After the first purification step, a noticeable drop in the signal intensity and a shift of the peak to 398.5 eV occurred, which can be attributed to the C=N-H group [44,47,48,49]. We hypothesised that, in addition to the removal of excess PVP, during the purification process, side reactions of pyrrolidone ring opening occurred. In the case of the conjugated sample, the intensity of the N 1s signal increased again and a double peak was detected. By deconvolution of the N 1s spectrum from the AuNP conjugated sample, shown in Figure 9b, we can extract three components. The dominant signal at 399.4 eV can still be attributed to the nitrogen in the pyrrolidone ring of PVP. Additionally, we can detect a side component of C=N- residues at 398.2 eV. Crucially, we detected a signal at 400.4 eV, which coincides with the signal of the N-terminal or peptide bonds [44,47,49].

Figure 10 shows the Au 4f XPS spectrum from the AuNP conjugated sample. It consists of the Au 4f_7/2_ and Au 4f_5/2_ doublet peaks. The Au 4f_7/2_ peak in Figure 10 was deconvoluted into two peaks, where the first Au 4f_7/2_ peak at 82.5 eV corresponds to the surface Au bonds, with a relative portion of 31%. The second Au 4f_7/2_ peak is at 83.6 eV and corresponds to the AuNP’s metallic core, with a relative portion of 69%.

Based on the XPS analyses, it was concluded that the specific binding of the capture proteins, SARS-CoV-2 S1 aa 1-681, takes place via a peptide bond formed between the citrate, which is coordinated to the AuNPs surface, and the free N-terminals of the protein. Vacant sites that were not occupied, either due to insufficient capture protein concentration or insufficient affinity, were filled by adding an excess of BSA. With this, the surface of the AuNP conjugate was “protected” from non-specific binding with other biomolecules. The proposed binding mechanism is presented in Figure 11.

### 3.4. Confirmation of AuNPs @ SARS-CoV-2 S1 Conjugate Functionality

The functionality of the prepared conjugates was confirmed within the scope of a broader clinical study approved by The Commission of the Republic of Slovenia for Medical Ethics, with Decision No. 0120-148/2021/3, and the Agency for Medicinal Products and Medical Devices of the Republic of Slovenia, with Decision No. 341-1/2021-10.

Within the study, a prototype LFIA test for the separate detection of IgA, IgG, and IgM SARS-CoV-2 S1 antibodies was evaluated using samples of human serum and nasal mucus. Testing showed that the separate detection of these antibodies is possible in patients who were infected with COVID-19 (Figure 12).

## 4. Conclusions

The following conclusions can be drawn from this study:The USP synthesis of gold (III) chloride at a Au concentration of 1 g/L in a precursor results in the formation of round AuNPs with a size of 50 nm ± 10 nm.Initial AuNPs have a low surface charge throughout the pH range from 2.4 mV at pH 2 to 5 mV at pH 11, which requires the use of stabilisers during USP synthesis for their long-term stability.The addition of SDS to the AuNP suspension results in the negative charge of their surface.The gel electrophoresis experiment confirmed the successful conjugation of AuNPs with SARS-CoV-2 S1 fragments indirectly, as it produced a longer gel trace. This is most likely due to the binding of additional SDS molecules to the SARS-CoV-2 S1 proteins in the AuNP conjugate.XPS analysis of the AuNP conjugate confirmed the presence of characteristic O 1s and N 1s peaks, nitrogen from PVP and carboxyl oxygen from citrate, on unpurified and purified samples.Comparison of the N 1s XPS spectra of unpurified and purified AuNPs compared to conjugated AuNPs revealed the formation of two obscured peaks and the presence of three sub-peaks: (i) a peak at 399.4 eV assigned to nitrogen in the pyrrolidone ring of PVP, (ii) a peak of C=N- at 398.2 eV, and (iii) a peak at 400.4 eV, which corresponds to the N-terminal signal or peptide bonds.The mechanism of AuNP conjugation with SARS-CoV-2 S1 is as follows: (i) During stabilisation, citrate ions bind to the surface of the AuNPs, which are preserved throughout the drying and cleaning processes. (ii) Raising the pH to 7.5 and adding proteins to the AuNP suspension allows them to bind to the free citrate carboxyl bond on the AuNPs’ surface through their free N-terminus. (iii) A peptide bond is formed.The functionality of conjugated SARS-CoV-2 S1 proteins was confirmed practically within the scope of a clinical study, where a prototype LFIA test based on the conjugate displayed the expected response.

## Figures and Tables

**Figure 1 nanomaterials-13-02160-f001:**
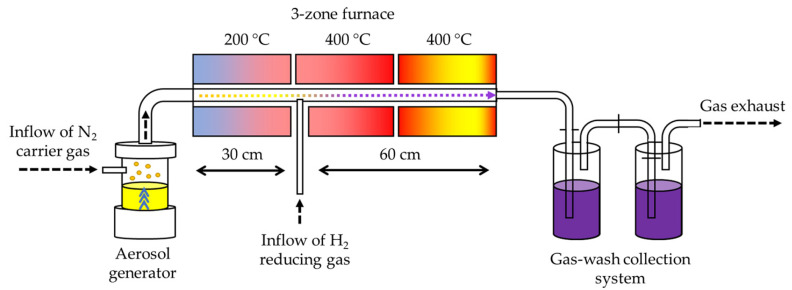
Schematic presentation of the custom USP device.

**Figure 2 nanomaterials-13-02160-f002:**
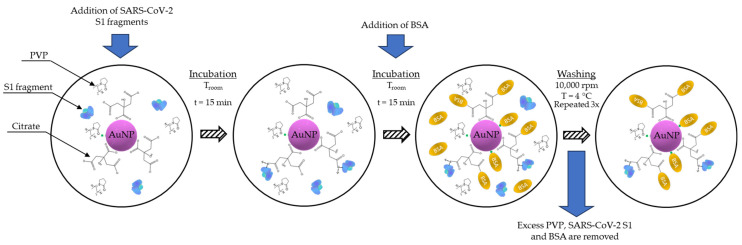
Conjugation process of AuNPs with SARS-CoV-2 S1 spike protein fragments.

**Figure 3 nanomaterials-13-02160-f003:**
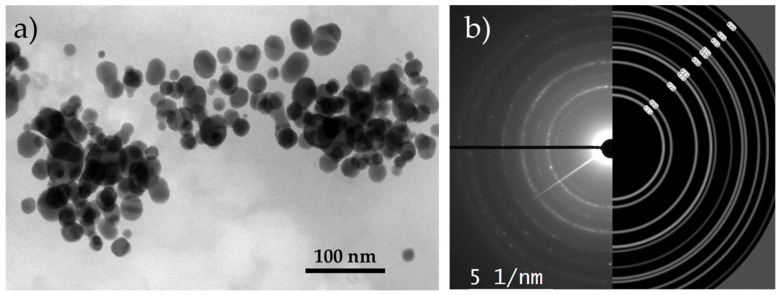
(**a**) TEM image of dried AuNPs, (**b**) experimental and simulated (Fm-3m, a = 4.0786 Å) diffraction pattern for AuNP.

**Figure 4 nanomaterials-13-02160-f004:**
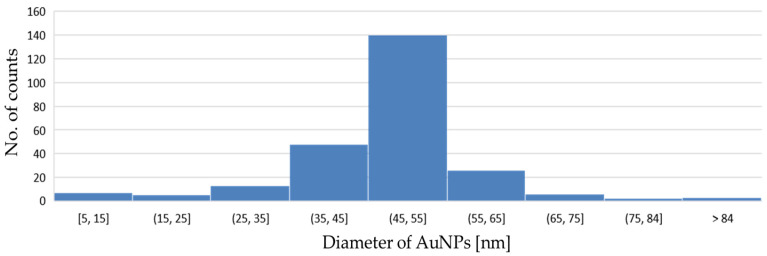
Size distribution of dried AuNPs.

**Figure 5 nanomaterials-13-02160-f005:**
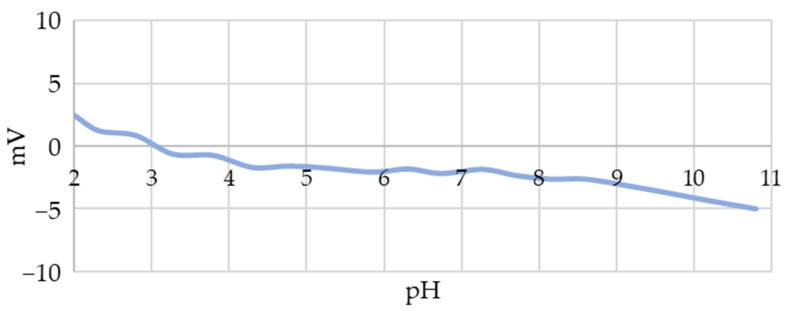
Zeta potential values of unstabilised AuNPs synthesised using USP as a function of pH.

**Figure 6 nanomaterials-13-02160-f006:**
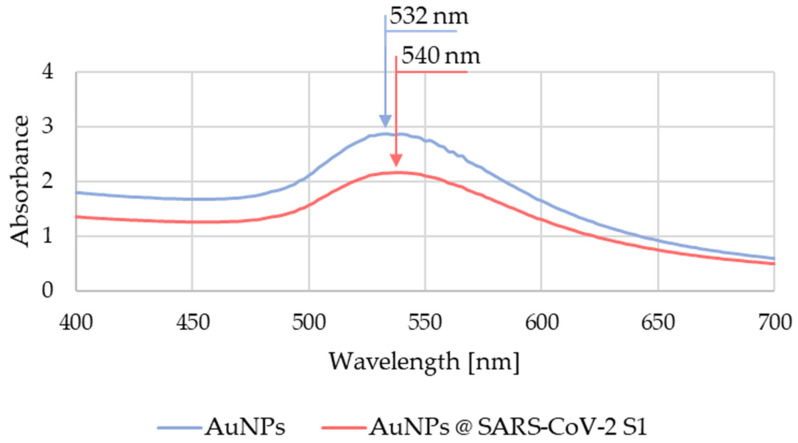
Measured UV-Vis absorption spectrum of the AuNP suspension before and after conjugation.

**Figure 7 nanomaterials-13-02160-f007:**
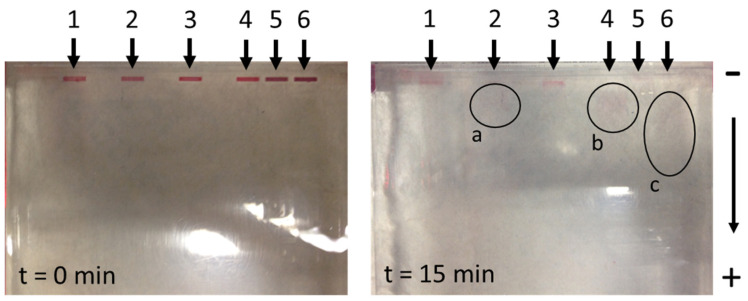
The travel results of unconjugated and conjugated AuNPs from different suspensions using the gel electrophoresis method (a—pathway representation of unconjugated AuNPs (modification with 1% SDS); b—showing the path of unconjugated AuNPs (modification with buffer excess and with 1% SDS); c—pathway representation of conjugated AuNPs (modification with 1% SDS).

**Figure 8 nanomaterials-13-02160-f008:**
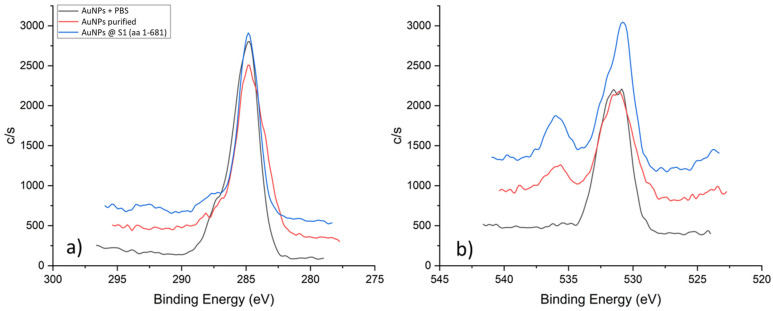
XPS spectra of C 1s (**a**) and O 1s (**b**) for different samples.

**Figure 9 nanomaterials-13-02160-f009:**
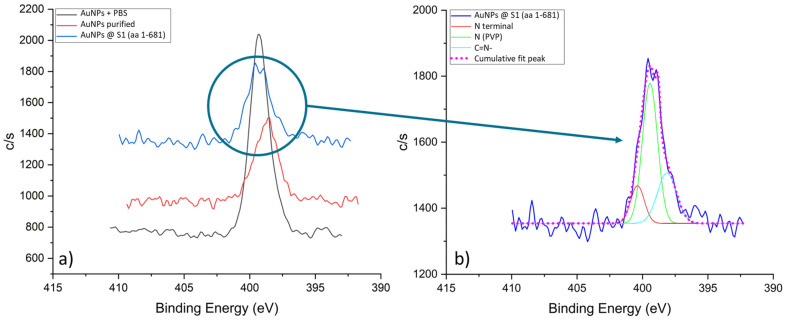
N 1s spectra for the AuNPs samples: (**a**) Stack of N 1s spectra, and (**b**) Fitted N 1s spectrum from the AuNP conjugated sample.

**Figure 10 nanomaterials-13-02160-f010:**
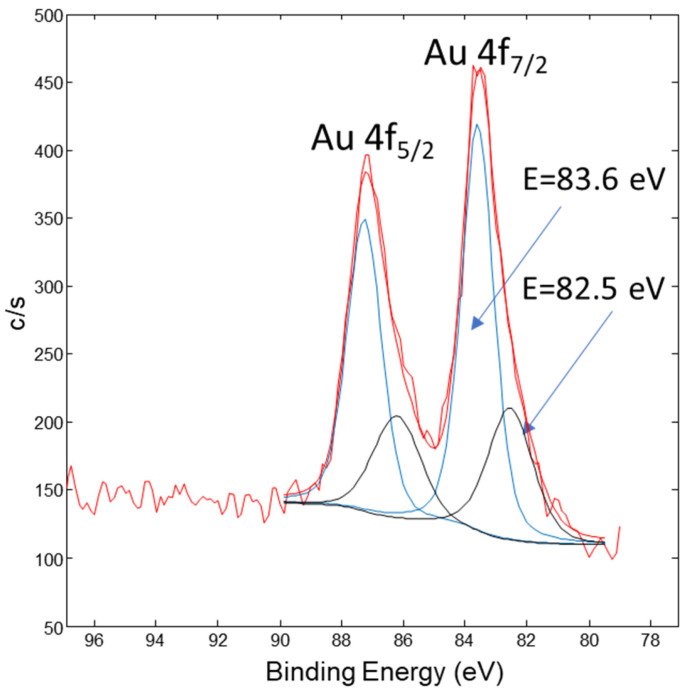
XPS spectrum of Au 4f from the AuNP conjugated sample. The red curve represents row data, the black peaks are related with AuNPs metallic core and the blue peaks are related with the surface Au-bonds.

**Figure 11 nanomaterials-13-02160-f011:**
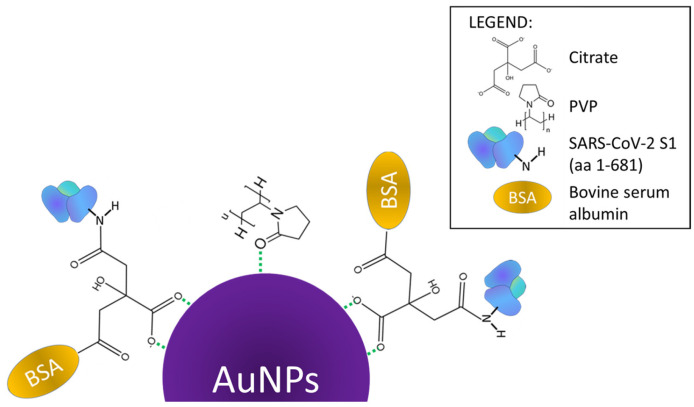
Conjugation mechanism of AuNPs with SARS-CoV-2 S1 aa 1-681 spike protein fragments.

**Figure 12 nanomaterials-13-02160-f012:**
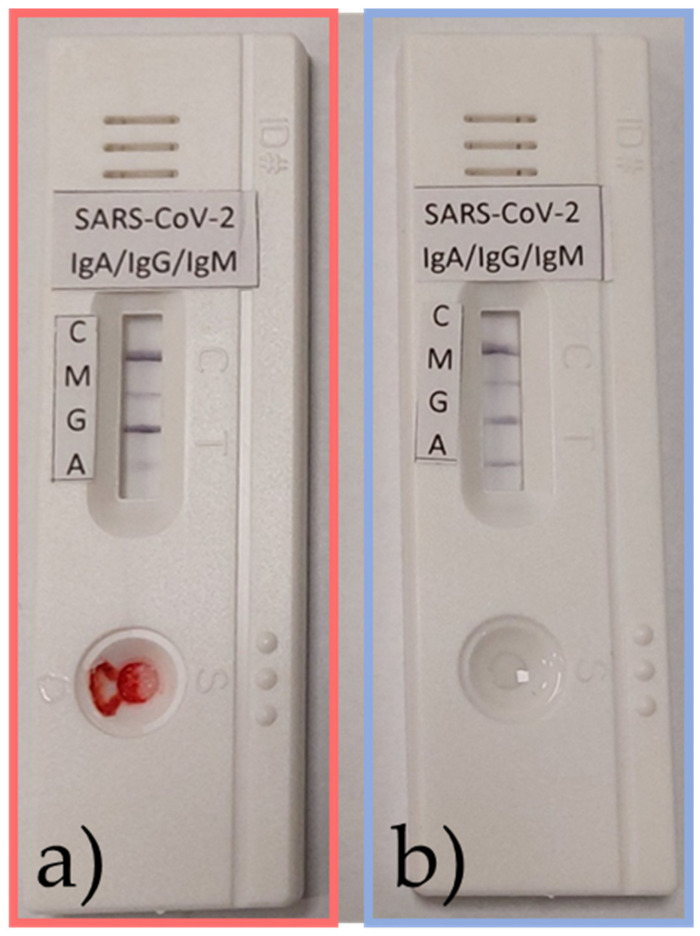
Samples from an active infection in a patient of (**a**) serum and (**b**) nasal mucus positive for IgA, IgG, and IgM SARS-CoV-2 S1 antibodies.

**Table 1 nanomaterials-13-02160-t001:** List of samples for XPS spectroscopy.

No.	Sample Composition
Sp1	AuNPs + (PBS/PVP/Citrate/Sucrose)
Sp2	Purified AuNPs + (PBS/PVP/Citrate/Sucrose)
Sp3	Purified AuNPs + (PBS/PVP/Citrate/Sucrose) conjugated with SARS-CoV-2 S1 (aa 1-681)

**Table 2 nanomaterials-13-02160-t002:** List of samples for gel electrophoresis testing.

No.	Sample	Addition of 1% SDS
1	AuNPs in PBS	
2	AuNPs in PBS	Yes
3	AuNPs in 5 mM Na_2_HPO_4_	
4	AuNPs in 5 mM Na_2_HPO_4_	Yes
5	AuNPs @ SARS-CoV-2 S1	
6	AuNPs @ SARS-CoV-2 S1	Yes

**Table 3 nanomaterials-13-02160-t003:** Surface chemical composition in at. % obtained with XPS analysis.

at. %								
Sample	C	N	O	Na	Si	P	Cl	Au
Sp1	67.0	3.1	24.5	2.6	1.0	0.4	1.3	0.2
Sp2	61.3	1.2	18.3	4.9	3.8	0.1	10.2	0.2
Sp3	51.0	2.7	19.9	10.1	1.3	3.1	11.3	0.3

## Data Availability

Data available on request from the corresponding author.

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
