# Peer review of "Study of Gold Nanoparticles Conjugated with SARS-CoV-2 S1 Spike Protein Fragments"

_nanomaterials, 2023, doi:10.3390/nano13152160_

Round 1

Reviewer 1 Report

This work reports the successful conjugation of SARS-CoV-2 S1 spike protein fragments with gold nanoparticles (AuNPs). The characterization of AuNPs and conjugation mechanism were also investigated by TEM, XPS and so forth. It is well planned, the results are properly described and discussed, and the conclusions are sound and supported by the data. Thus, I recommend this manuscript for publication in nanomaterials after some major revisions towards the following points:

1.     The authors should rephrase the abstract such that it better reflects the contents.

2.     To identify the successful synthesis of AuNPs, the UV spectrum of AuNPs should be added.

3.     From Figure 3a, the agglomeration of gold nanoparticles is quite obvious, so whether the author considers the influencing factors, including the content of HAuCl4 , H2 and PVP, the reactive time…...

4.     In Figure 10, the symbol for the AuNPs was written wrong.

5.     Overall discussion conjugation mechanism is ok however it should be improved.

6.     There are many methods for synthesizing AuNPs, and what are the advantages of using this method by the author.

7.     For the references, it is recommended that the author reduce to about 50.

Author Response

Review 1:

This work reports the successful conjugation of SARS-CoV-2 S1 spike protein fragments with gold nanoparticles (AuNPs). The characterization of AuNPs and conjugation mechanism were also investigated by TEM, XPS and so forth. It is well planned, the results are properly described and discussed, and the conclusions are sound and supported by the data. Thus, I recommend this manuscript for publication in nanomaterials after some major revisions towards the following points:

  1. The authors should rephrase the abstract such that it better reflects the contents.

The abstract has been amended.

  1. To identify the successful synthesis of AuNPs, the UV spectrum of AuNPs should be added.

We have added a new chapter with UV-Vis measurements of AuNPs before and after conjugation.

  1. From Figure 3a, the agglomeration of gold nanoparticles is quite obvious, so whether the author considers the influencing factors, including the content of HAuCl4 , H2 and PVP, the reactive time…...

Figure 3a, represents gold nanoparticles specially prepared with anhydrous ethanol, so that a drop was added onto a TEM grid and dried. Since these are not the typical environmental factors under which the gold nanoparticles are uses, this also must be accounted for, when analysing the image. While some agglomeration or overlapping is visible, the majority of the gold nanoparticles are monodispersed. As can be seen from the figure 3a between the gold nanoparticles (the nanoparticles remained pink/purple after drying, which was further indirect evidence that they did not agglomerate - instead they would have a grey colour).

  1. In Figure 10, the symbol for the AuNPs was written wrong.

Thank you for pointing out our mistake. It has been fixed. 

  1. Overall discussion conjugation mechanism is ok however it should be improved.

We have added new results which deal with 2.7. Preparation of LFIA test strips and 3.3. Confirmation of AuNPs @ SARS-CoV-2 S1 conjugate functionality, thereby further confirming that conjugation had occurred.

  1. There are many methods for synthesizing AuNPs, and what are the advantages of using this method by the author.

From a production point of view, the main advantage offered by USP is the continuity and high rate of achieving uniform nanoparticle size. Increasing the capacity (g/h) of the USP device is also quite simple by increasing the diameter, length or number of reaction tubes. In the context of conjugation, USP offers a relatively easy possibility to synthesize gold nanoparticles using steric stabilizers in a collection system, thereby preventing their agglomeration.

We have expanded the explanation in the article – Lines 67 to 73.

  1. For the references, it is recommended that the author reduce to about 50.

As per your suggestion, we have removed the following references:

Schuetze, B.; Mayer, C.; Loza, K.; Gocyla, M.; Heggen, M.; Epple, M. Conjugation of thiol-terminated molecules to ultrasmall 2 nm-gold nanoparticles leads to remarkably complex 1H-NMR spectra. J. Mater. Chem. B 2016, 4, 2179–2189. 10.1039/c5tb02443a.

Brocchini, S.; Badescu, G.; Bryant, P.; Swierkosz, J.; Khayrzad, F.; Pawlisz, E.; Farys, M.; Cong, Y.; Rumpf, N.; Godwin, A. Article A new reagent for stable thiol specific conjugation. 2013.

Patra, M.; Eichenberger, L.S.; Fischer, G.; Holland, J.P. Photochemical Conjugation and One-Pot Radiolabelling of Antibodies for Immuno-PET. Angew. Chemie - Int. Ed. 2019, 58, 1928–1933. 10.1002/anie.201813287.

Pingale, S.S. Molecular electrostatic potential for exploring π-conjugation: A density-functional investigation. Phys. Chem. Chem. Phys. 2011, 13, 15158–15165. 10.1039/c1cp20071b.

Jazayeri, M.H.; Amani, H.; Pourfatollah, A.A.; Pazoki-Toroudi, H.; Sedighimoghaddam, B. Various methods of gold nanoparticles (GNPs) conjugation to antibodies. Sens. Bio-Sensing Res. 2016, 9, 17–22. 10.1016/j.sbsr.2016.04.002.

Bachelet, M. Design of pH-responsive gold nanoparticles in oncology. Mater. Sci. Technol. (United Kingdom) 2016, 32, 794–804. 10.1179/1743284715Y.0000000090.

Farooq, M.U.; Novosad, V.; Rozhkova, E.A.; Wali, H.; Ali, A.; Fateh, A.A.; Neogi, P.B.; Neogi, A.; Wang, Z. Gold Nanoparticles-enabled Efficient Dual Delivery of Anticancer Therapeutics to HeLa Cells. Sci. Rep. 2018, 8, 1–12. 10.1038/s41598-018-21331-y.

Luo, D.; Wang, X.; Burda, C.; Basilion, J.P. Recent development of gold nanoparticles as contrast agents for cancer diagnosis. Cancers (Basel). 2021, 13. 10.3390/cancers13081825.

Srivastava, M.; Nirala, N.R.; Srivastava, S.K.; Prakash, R. A comparative Study of Aptasensor Vs Immunosensor for Label-Free PSA Cancer Detection on GQDs-AuNRs Modified Screen-Printed Electrodes. Sci. Rep. 2018, 8, 1–11. 10.1038/s41598-018-19733-z.

Lymberis, A. Smart wearables for remote health monitoring, from prevention to rehabilitation: Current RandD, future challenges. Proc. IEEE/EMBS Reg. 8 Int. Conf. Inf. Technol. Appl. Biomed. ITAB 2003, 2003-Janua, 272–275. 10.1109/ITAB.2003.1222530.

Aldewachi, H.; Chalati, T.; Woodroofe, M.N.; Bricklebank, N.; Sharrack, B.; Gardiner, P. Gold nanoparticle-based colorimetric biosensors. Nanoscale 2018. 10.1039/c7nr06367a.

Vashist, S.K.; Luong, J.H.T. Immunoassays: An overview; Elsevier Inc., 2018; ISBN 9780128117620. 10.1016/B978-0-12-811762-0.00001-3.

Sengani, M.; Grumezescu, A.M.; Rajeswari, V.D. Recent trends and methodologies in gold nanoparticle synthesis – A prospective review on drug delivery aspect. OpenNano 2017, 2, 37–46. 10.1016/j.onano.2017.07.001.

Wen, T.; Huang, C.; Shi, F.J.; Zeng, X.Y.; Lu, T.; Ding, S.N.; Jiao, Y.J. Development of a lateral flow immunoassay strip for rapid detection of IgG antibody against SARS-CoV-2 virus. Analyst 2020, 145, 5345–5352. 10.1039/d0an00629g.

Tomko, J.; O’Malley, S.M.; Trout, C.; Naddeo, J.J.; Jimenez, R.; Griepenburg, J.C.; Soliman, W.; Bubb, D.M. Cavitation bubble dynamics and nanoparticle size distributions in laser ablation in liquids. Colloids Surfaces A Physicochem. Eng. Asp. 2017, 522, 368–372. 10.1016/j.colsurfa.2017.03.030.

Kim, D.S.; Kim, Y.T.; Hong, S.B.; Kim, J.; Huh, N.S.; Lee, M.K.; Lee, S.J.; Kim, B. Il; Kim, I.S.; Huh, Y.S.; et al. Development of lateral flow assay based on size-controlled gold nanoparticles for detection of hepatitis B surface antigen. Sensors (Switzerland) 2016, 16. 10.3390/s16122154.

Shariq, M.; Majerič, P.; Friedrich, B.; Budic, B.; Jenko, D.; Dixit, A.R.; Rudolf, R. Application of Gold(III) Acetate as a New Precursor for the Synthesis of Gold Nanoparticles in PEG Through Ultrasonic Spray Pyrolysis. J. Clust. Sci. 2017, 28, 1647–1665. 10.1007/s10876-017-1178-0.

Rudolf, R.; Majerič, P.; Štager, V.; Albreht, B. Process for the production of gold nanoparticles by modified ultrasonic spray pyrolysis: patent application no. P-202000079. Ljubljana: Office of the Republic of Slovenia for Intellectual Property 2020.

Amis, T.M.; Renukuntla, J.; Bolla, P.K.; Clark, B.A. Selection of cryoprotectant in lyophilization of progesterone-loaded stearic acid solid lipid nanoparticles. Pharmaceutics 2020, 12, 1–15. 10.3390/pharmaceutics12090892.

Reviewer 2 Report

 This study reports on the conjugation of SARS-CoV-2 S1 spike protein fragments with gold nanoparticles. X-ray photoelectron spectroscopy (XPS) is used expertly to evaluate and describe this conjugation mechanism. However, the lack of information regarding the capacity of these engineered nanoparticles to recognize the cellular target raises serious questions regarding their potential to be used as biosensors.

 Specific comments.

 - Monitoring the C 1s, N 1s, O 1s, and Au 4f spectra is a key to understanding the conjugation mechanism between AuNPs and SARS-CoV.2 S1 fragments but does not prove the binding of SARS-CoV-2 S1 spike protein to AuNPs

- Additional experiments are needed to provide evidence that SARS-CoV-2 S1 spike protein fragments conjugated with AuNPs bind to its cognate receptor,angiotensin-converting enzyme 2(ACE2). Binding does not mean that SARS-CoV-2 S1 spike protein is still active and folded in the right conformation.

- The statement that vacant sites are filled(Figure 10) by adding an excess of BSA is highly speculative and not supported by experimental evidence. Indeed XPS measures the binding via a peptide bond formed between citrate and the free N-terminals of the protein and XPS signal will be similar whatever the protein being used.

Author Response

Review 2:

This study reports on the conjugation of SARS-CoV-2 S1 spike protein fragments with gold nanoparticles. X-ray photoelectron spectroscopy (XPS) is used expertly to evaluate and describe this conjugation mechanism. However, the lack of information regarding the capacity of these engineered nanoparticles to recognize the cellular target raises serious questions regarding their potential to be used as biosensors.

 Specific comments.

 - Monitoring the C 1s, N 1s, O 1s, and Au 4f spectra is a key to understanding the conjugation mechanism between AuNPs and SARS-CoV.2 S1 fragments but does not prove the binding of SARS-CoV-2 S1 spike protein to AuNPs

- Additional experiments are needed to provide evidence that SARS-CoV-2 S1 spike protein fragments conjugated with AuNPs bind to its cognate receptor, angiotensin-converting enzyme 2(ACE2). Binding does not mean that SARS-CoV-2 S1 spike protein is still active and folded in the right conformation.

Thanks for the comments. The common point you raised is that our results need further confirmation of their functionality.

Based on your requirements, we have added results that confirm their functionality, namely in the following chapters: 2.7. Preparation of LFIA test strips and 3.3. Confirmation of AuNPs @ SARS-CoV-2 S1 conjugate functionality.

As part of these additional studies, we successfully used S1 fragments to capture S1 antibodies and confirm their presence in human serum and nasal mucus samples.

- The statement that vacant sites are filled (Figure 10) by adding an excess of BSA is highly speculative and not supported by experimental evidence. Indeed XPS measures the binding via a peptide bond formed between citrate and the free N-terminals of the protein and XPS signal will be similar whatever the protein being used.

Using excess BSA is quite routine in the context of preparing conjugates for use in LFIA and similar tests [1]. As you pointed out the binding of BSA has been well explored and is typically via a peptide bond, as confirmed by our results. A deviation from this would indicate, that a different binding mechanism occurred for the SARS-CoV-2 S1. As there was no evidence to support a different binding, we can conclude, that the binding occurred via a peptide bond.

  1. Parolo, C.; Sena-Torralba, A.; Bergua, J.F.; Calucho, E.; Fuentes-Chust, C.; Hu, L.; Rivas, L.; Álvarez-Diduk, R.; Nguyen, E.P.; Cinti, S.; et al. Tutorial: design and fabrication of nanoparticle-based lateral-flow immunoassays. Nat. Protoc. 2020, 15, 3788–3816. 10.1038/s41596-020-0357-x.

Round 2

Reviewer 1 Report

Accept.

Reviewer 2 Report

Remarks and concerns have been addressed convincingly and thoughtfully. The new data described in the amended chapter 3.3 demonstrate the functionality of the prepared conjugates and improve significantly the impact of the manuscript